# Efficacy of abiraterone acetate for high-risk hormone-naïve metastatic prostate cancer: A comparison with combined androgen blockade therapy with bicalutamide and androgen deprivation therapy alone

**Kent Kanao**[ORCID]*, **Takayuki Takahashi, Yuta Umezawa, Takashi Okabe, Go Kaneko, Suguru Shirotake, Koshiro Nishimoto, Masafumi Oyama**

Department of Uro-Oncology, Saitama Medical University International Medical Center, Hidaka-City, Saitama-Pref, Japan

* kanaok@saitama-med.ac.jp

**Data Availability Statement:** All relevant data are within the manuscript.

## Abstract

### Background

The treatment landscape for men with metastatic hormone-naïve prostate cancer (mHNPC) has dramatically changed with the approval of next-generation anti-androgen drugs. We compared the treatment efficacy of abiraterone with that of combined androgen blockade (CAB) therapy and androgen deprivation therapy (ADT) alone in men with high-risk mHNPC.

### Methods

In total, 146 Japanese men with high-risk mHNPC were retrospectively analyzed. As initial hormonal therapy, 30, 83, and 33 men were treated with ADT plus abiraterone (ABI group), ADT plus bicalutamide (CAB group), and ADT alone (ADT group), respectively. Treatment efficacy was compared using time to castration resistance (TTCR) and prostate-specific antigen (PSA) response among the groups. Propensity score matching analysis was also performed to adjust for baseline differences.

### Results

The median (95% confidence interval [CI]) TTCR in the ABI, CAB, and ADT groups were not reached, 10.7 (7.6–13.8) months and 11.0 (7.9–12.4) months, respectively, and it was significantly longer in the ABI group than in the other groups (p = 0.0012, p = 0.0008). In propensity score matching analysis, the median TTCR was also significantly longer in the ABI group than in the other groups (hazard ratio [HR], 0.47; 95% CI, 0.22–0.98; p = 0.010; HR, 0.32; 95% CI, 0.12–0.85; p = 0.004). The number of men who achieved PSA levels ≤0.2 ng/mL after propensity score matching were significantly higher in the ABI group than in the other groups.

**Funding:** Kent Kanao was supported by JSPS KAKENHI Grant Number JP 18K07210. The funders had no role in study design, data collection and analysis, decision to publish, or preparation of the manuscript.

**Competing interests:** The authors have declared that no competing interests exist.

## Conclusions

Our results provide important evidence regarding the superiority of abiraterone over CAB therapy and ADT alone for initial treatment for men with newly diagnosed mHNPC.

## Introduction

In the past few years, the treatment landscape for men with metastatic castration-sensitive (mCSPC) or hormone-naïve prostate cancer (mHNPC) has dramatically changed based on results of several large, randomized, phase 3 clinical trials [1–6]. These trials have shown longer survival in men with metastatic prostate cancer when androgen-deprivation therapy (ADT) was combined with abiraterone acetate (hereafter referred to as abiraterone), enzalutamide, apalutamide, or docetaxel at the time of initial ADT administration.

In Japan, abiraterone was approved for men with high-risk mHNPC in 2018 based on the findings of the LATITUDE study, which showed a significant increase in the overall survival (OS) and radiographic progression-free survival (rPFS) in men with newly diagnosed high-risk mCSPC. However, combined androgen blockade (CAB) therapy with bicalutamide (80 mg daily) had been generally used in Japanese men with mHNPC based on the results of a large multicenter randomized controlled trial [7]. Therefore, a comparison of abiraterone and bicalutamide treatment in men with mHNPC may provide information regarding the selection of initial treatment because the control arm in the LATITUDE study used ADT alone [4]. Further, to our knowledge, few study has directly compared the treatment efficacy of abiraterone and bicalutamide in men with mHNPC, and no study has compared the treatment efficacy of them including time to castration resistance and PSA response rate using propensity score matching.

In this study, we compared the treatment efficacy of abiraterone with that of bicalutamide and ADT alone in Japanese men with newly diagnosed high-risk mHNPC.

## Materials and methods

### Patients

In this retrospective single institutional study, we identified a total of 146 men with newly diagnosed mHNPC between 2010 and 2020 who met the high-risk criteria according to the LATITUDE study [4]. Patients initially treated with radical prostatectomy or radiotherapy were excluded from this study. Clinicopathological data on age and prostate-specific antigen (PSA) level at diagnosis, clinical stage, Gleason score at diagnosis, sites of metastasis, and number of bone metastasis were obtained from the medical records of each patient. The PSA level was measured every 4–12 weeks, while radiological examinations were performed at the physician's discretion. Ethical approval for this study was granted by the ethics committee of Saitama Medical University International Medical Center (19–237). Informed consent was not required as this was a retrospective study and data were anonymized.

### Treatment and efficacy assessment

As initial hormonal therapy, 30 men were treated with ADT plus abiraterone (1000 mg daily) plus prednisolone (5 mg daily) (ABI group), 83 were treated with ADT plus 80 mg bicalutamide (CAB group), and 33 men were treated with ADT alone (ADT group). Men in the ABI group were administered abiraterone within 3 months of initial ADT administration, and

those in the CAB group were administered bicalutamide within 1 month before initial ADT administration. Bilateral orchiectomy, luteinizing hormone-releasing hormone agonist, or gonadotropin-releasing hormone (GnRH) antagonist was used as ADT. However, in the ADT group, only GnRH antagonist (degarelix) was used as ADT.

Treatment efficacy was compared using time to castration resistance (TTCR) and PSA response among the ABI, CAB, and ADT groups. TTCR was calculated from the initiation of initial ADT until the first development of castration resistance. Castration resistance was defined according to the Prostate Cancer Clinical Trials Working Group 3 [8].

## Statistical analysis

The baseline characteristics of the three groups were compared using the chi-square and Kruskal–Wallis tests for categorical and continuous variables, respectively. The Kaplan–Meier method was used to estimate TTCR, and differences among the groups were evaluated using the log-rank test and Kruskal-Wallis test.

Furthermore, propensity score matching analysis was performed to adjust for baseline differences. Logistic regression models were used to calculate the propensity scores; the following parameters were considered: age, initial PSA levels, Gleason score, T stage (T4 or ≤T3), N stage (N1 of N0), and visceral metastases (yes/no). Matches were created using the most common methods, the nearest neighbor method, using one-to-one matching with a caliper width of 0.2. Hazard ratios (HR) and odds ratios (OR) were estimated using inverse probability of treatment weighting.

A p value of <0.05 was considered statistically significant. All statistical analyses were performed using the statistical software R version 3.3.1. (R Core Team, 2020) and the package "Matching" and "Survival".

## Results

The median (range) age and PSA level at diagnosis were 70 (47–91) years and 415 (6.5–23022) ng/mL. The number of men with metastasis of the extra-regional lymph node, bone, and viscera at diagnosis was 4, 137, and 34, respectively. The characteristics and differences of the three groups are summarized in Table 1. The median PSA level at diagnosis was higher in the ABI group than in the CAB and ADT group, but there were no significant differences with regard to men characteristics among the groups.

The median (range) follow-up period in the ABI, CAB, and ADT groups was 12.6 (3–90), 9.6 (1–115), and 9.6 (3–52) months, respectively. In total, 9, 69, and 27 men in the ABI, CAB, and ADT groups, respectively, progressed to castration-resistant prostate cancer (CRPC) during the follow-up period. The median follow-up period of ABI group was a little longer than CAB and ADT groups, but we followed these patients uniformly though time of treatment initiation varied. Thirteen patients were referred to another hospital during the treatment, 6 in the ABI and 7 in the CAB groups, and 3 patients died of other causes. The number of men with a ≥90% decrease in the PSA level was 30 (100%), 75 (90%), and 28 (85%) in the ABI, CAB, and ADT groups, respectively. Further, 19 (63%), 21 (25%), and 1 (3%) in the ABI, CAB, and ADT groups, respectively, achieved a PSA level of ≤0.2 ng/mL (Table 2). The median (95% confidence interval [CI]) TTCR in the ABI, CAB, and ADT groups were not reached, 10.7 (7.6–13.8) months and 11.0 (7.9–12.4) months, respectively, and it was significantly longer in the ABI group than in the CAB and ADT groups (p = 0.0012 and p = 0.0008, respectively; Fig 1). However, there was no difference of the median TTCR between the CAB and ADT groups (p = 0.546).

**Table 1. Baseline characteristics of patients.**

| | Abiraterone (ABI; n=30) | CAB with bicalutamide (CAB; n=83) | ADT alone (ADT; n=33) | p value |
|---|---|---|---|---|
| Median age at diagnosis (range), years | 68 (54-82) | 72 (49-91) | 71 (47-84) | 0.440 |
| Median PSA at diagnosis (range), ng/mL | 531 (7.7-23022) | 381 (8.6-8855) | 361 (6.5-7155) | 0.054 |
| Clinical T stage, n (%) | | | | 0.211 |
| ≤T3 | 23 (77) | 64 (77) | 30 (91) | |
| T4 | 7 (23) | 19 (23) | 3 (9) | |
| Gleason score, n (%) | | | | 0.111 |
| 4+4 | 9 (30) | 41 (49) | 12 (36) | |
| 4+5 | 16 (53) | 19 (23) | 11 (33) | |
| 5+4 | 4 (13) | 16 (19) | 7 (21) | |
| 5+5 | 1 (3) | 7 (8) | 3 (9) | |
| Bone metastasis, n (%) | | | | 0.323 |
| No | 3 (10) | 3 (4) | 1 (3) | |
| Yes | 27 (90) | 80 (96) | 32 (97) | |
| Regional node metastasis, n (%) | | | | 0.937 |
| No | 13 (43) | 38 (46) | 14 (42) | |
| Yes | 17 (57) | 45 (54) | 19 (58) | |
| Visceral metastasis, n (%) | | | | 0.802 |
| No | 22 (73) | 62 (75) | 28 (85) | |
| Yes | 8 (27) | 21 (25) | 5 (15) | |

In propensity score matching analysis, the median TTCR was also significantly longer in the ABI group than the CAB and ADT groups (hazard ratio [HR], 0.47; 95% CI, 0.22–0.98; p = 0.010 and HR, 0.32; 95% CI, 0.12–0.85; p = 0.004, respectively). However, there was no difference in the median TTCR between the CAB and ADT groups (HR, 0.83; 95% CI, 0.53–1.3; p = 0.40). Fig 2 shows a comparison of the Kaplan–Meier curves between the three groups after propensity score matching.

A comparison of the percentage of men who achieved PSA levels ≤0.2 between the three groups after propensity score matching is shown in Fig 3. The percentages of the ABI group were significantly higher than those of the CAB and ADT groups (OR, 8.2; 95% CI, 2.8–24; p<0.001 and OR, 114.2; 95% CI, 13–1040; p<0.001, respectively). The percentage of men in the CAB group was also significantly higher than that of the ADT group (OR, 11.4; 95% CI, 1.4–90; p = 0.023).

Adverse events with dose reduction or interruption occurred in 7 patients, all in the abiraterone group. Most of them were ALT/AST increase. All patients who interrupted dose

**Table 2. Treatment outcomes of patients.**

| Variables | Abiraterone (ABI; n=30) | CAB with bicalutamide (CAB; n=83) | ADT alone (ADT; n=33) | p value |
|---|---|---|---|---|
| Median observation time, months (range) | 12.6 (3-90) | 9.6 (1-115) | 9.6 (3-52) | 0.458 |
| Patients progressed to CRPC, n (%) | 9 (30) | 69 (83) | 27 (82) | <0.001 |
| Median TTCR, months (95% CI) | not reached | 10.7 (7.6-13.8) | 11.0 (7.9-12.4) | 0.002 |
| PSA nadir, median (range), ng/mL | 0.094 (<0.008-9.7) | 1.21 (<0.008-2672) | 2.818 (0.041-234) | <0.001 |
| PSA decline, n (%) | | | | |
| Any | 30 (100) | 81 (98) | 33 (100) | 0.4632 |
| ≥90% | 30 (100) | 75 (90) | 28 (85) | 0.1015 |
| Patients reaching PSA ≤0.2ng/mL, n (%) | 19 (63) | 21 (25) | 1 (3) | <0.001 |

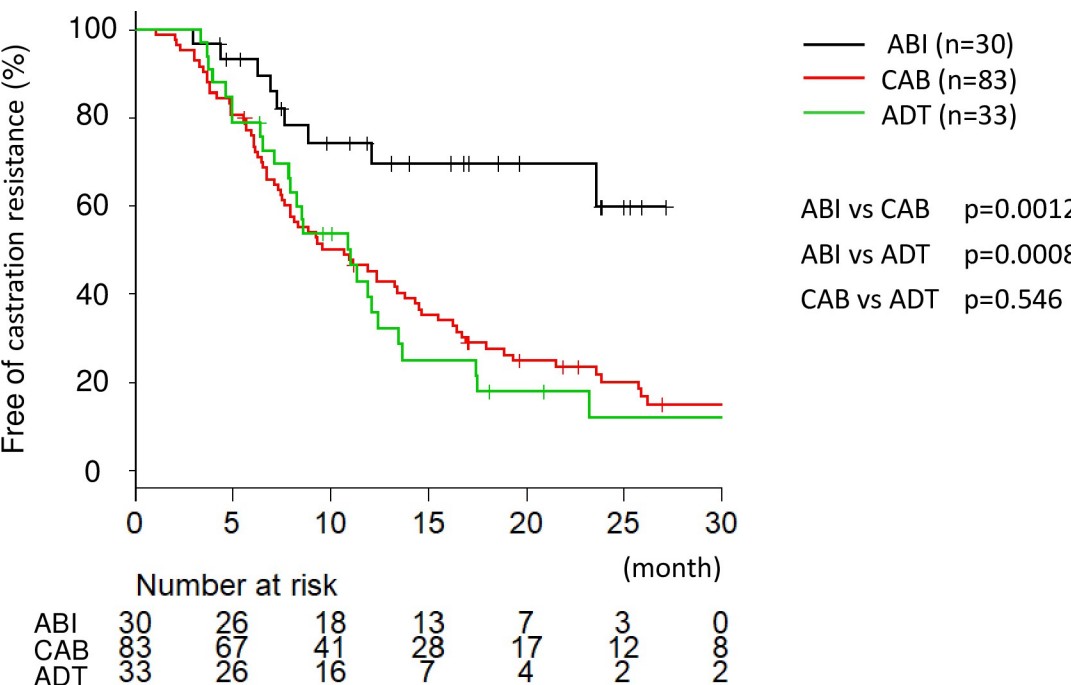

**Fig 1. Kaplan–Meier curve comparing time to castration resistance (TTCR) among the three groups.** The median TTCR in the androgen deprivation therapy (ADT) plus abiraterone (ABI group) was significantly longer than that in the combined androgen blockade (CAB) and ADT alone groups. However, there was no difference in the median TTCR between the CAB and ADT groups.

resumed abiraterone and none discontinued. Details of adverse events of abiraterone group were shown in Table 3.

## Discussion

In this study, we showed that the median TTCR was significantly longer in men with newly diagnosed high-risk mHNPC treated with abiraterone than in those treated with bicalutamide.

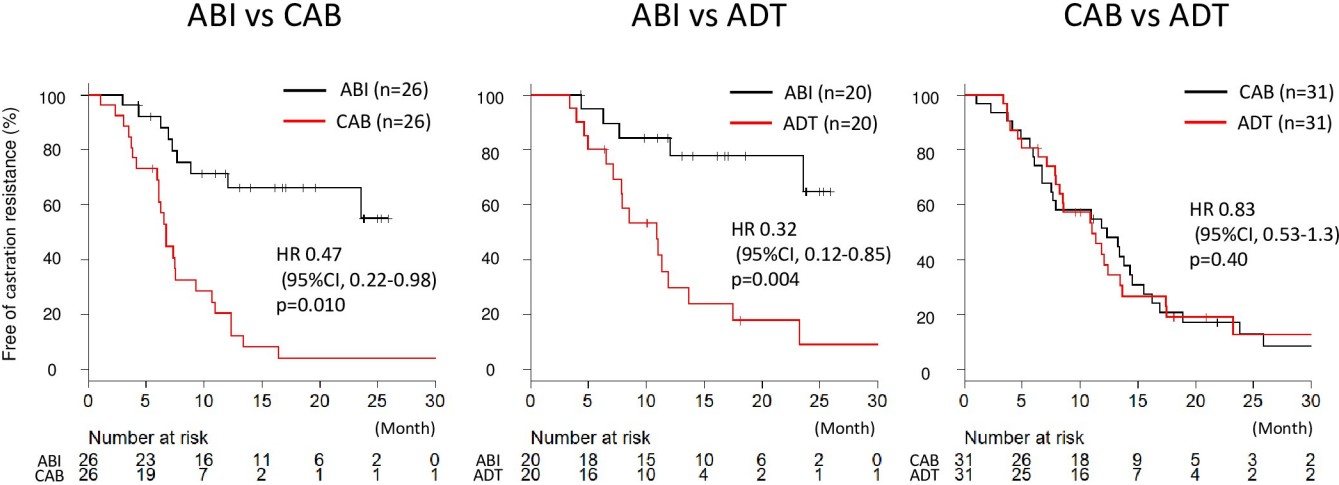

**Fig 2. Kaplan–Meier curve after propensity score matching.** The median time to castration resistance (TTCR) in the androgen deprivation therapy (ADT) plus abiraterone (ABI group) was significantly longer than that in the combined androgen blockade (CAB) and ADT alone groups. However, there was no difference of the median TTCR between the CAB and ADT groups.

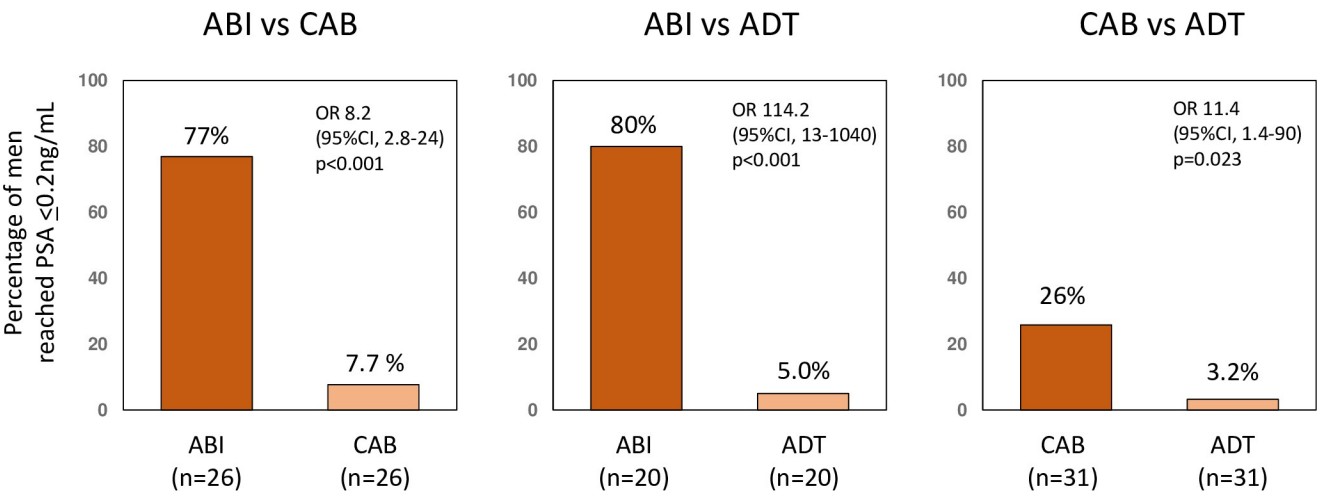

**Fig 3. Percentage of men achieved a prostate-specific antigen (PSA) level of ≤0.2 after propensity score matching.** The percentage of men achieved a prostate-specific antigen (PSA) level of ≤0.2 was significantly higher than in the androgen deprivation therapy (ADT) plus abiraterone (ABI group) than in the combined androgen blockade (CAB) and ADT groups.

The effect of first-generation antiandrogens, such as bicalutamide, nilutamide, or flutamide, in addition to ADT has been widely debated because of conflicting data from individual clinical trials as well as tolerability and cost issues. In 2009, Akaza et al. in a double-blind randomized controlled study demonstrated that ADT plus 80 mg bicalutamide had a statistically significant OS benefit in men with locally advanced or metastatic prostate cancer compared with ADT alone during long-term follow-up [7]. Based on the study by Akaza et al., ADT plus 80 mg bicalutamide has been the standard initial treatment for metastatic prostate cancer in Japan [9]. Therefore, our results indicating superiority of abiraterone over CAB therapy provide an important evidence regarding the initial treatment in men with high-risk mHNPC. Furthermore, we showed that bicalutamide did not have any additional benefit over ADT. This result is consistent with the that reported in the study by Akaza et al., which showed that the difference in OS was greater in men with stage C/D1 disease; however, there was no difference among men with stage D2 disease [7]. Therefore, it is considered that bicalutamide does not provide any additional effect over ADT for men with high-risk mHNPC.

In this study, we used TTCR to evaluate the efficacy of abiraterone. OS is the most objective and convenient measure of meaningful clinical efficacy of investigational drugs. Moreover, rPFS has also been used as a surrogate for OS in several clinical trials. However, it is difficult to use OS for comparing treatments in cases where the control was historical and affected by historical differences in subsequent therapy. It is also difficult to use rPFS in retrospective studies

**Table 3. Adverse events of abiraterone leading to treatment dose reduction and interruption.**

|  | Adverse event | Grade | Time to onset | Response |
|---|---|---|---|---|
| #1 | ALT/AST increased | G3 | 1.5 mo. | Dose Reduction |
| #2 | ALT/AST increased | G2 | 1 mo. | Dose Interruption |
| #3 | ALT/AST increased | G1 | 2.5 mo. | Dose Reduction |
| #4 | ALT/AST increased | G1 | 0.5 mo. | Dose Interruption |
| #5 | ALT/AST increased | G1 | 2 mo. | Dose Interruption |
| #6 | Hyperglycemia | G2 | 4 mo. | Dose Interruption |
| #7 | Nausea | G1 | 1 mo. | Dose Reduction |

such as our study as most men did not plan scheduled radiographic examination to evaluate disease progression before advancement to CRPC. Recently, several studies have indicated that TTCR can be a good surrogate endpoint of OS in men with metastatic prostate cancer [10, 11]. Frees et al. showed that TTCR is a valid surrogate of OS and concluded that it should be prolonged as much as possible [10]. Miyake et al. also demonstrated that men with mCSPC who had a longer TTCR are likely to achieve a more favorable OS [11]. Therefore, we used TTCR as an endpoint to compare the treatment efficacy in men with mHNPC and we concluded that abiraterone may have a significant OS benefit compared with CAB therapy and ADT alone in men with newly diagnosed high-risk mHNPC.

We also compared the percentage of men who achieved PSA levels ≤0.2 ng/mL between the three groups after propensity score matching; the percentage of men who achieved PSA levels ≤0.2 was significantly higher in the ABI group than in the CAB and ADT groups. A decline in PSA levels is a well-established predictor of prognosis in men with mHNPC treated with primary ADT [12–14]. Harshman et al. reported that a PSA level of ≤0.2 ng/mL at 7 months is a prognostic factor for longer OS with ADT for mHNPC irrespective of docetaxel administration [15]. Recently, Matsubara et al. in a post hoc analysis of the LATITUDE study showed that men treated with abiraterone plus ADT were 6.1 times more likely to achieve a PSA level of <0.2 ng/mL than those receiving placebo plus ADT (55% vs. 9%) [16]. Our result is consistent with that reported in the study Matsubara et al. and suggests that newly diagnosed high-risk mHNPC men treated with abiraterone had a significantly better OS than men treated with CAB therapy and ADT.

In this study, adverse events with dose reduction or interruption occurred in 7 patients, all in the abiraterone group. However, all patients who interrupted dose resumed abiraterone and none discontinued. Therefore, men with high-risk mHNPC treated with abiraterone had a manageable safety profile.

This study had some limitations. This study is a retrospective single-institutional study with a relatively small cohort and short follow-up period. As mentioned above, this study evaluated TTCR and PSA response as the study endpoint as a surrogate for OS. However, we believe our results indicate superiority of abiraterone over CAB therapy, providing important evidence for initial treatment in men with mHNPC.

## Conclusion

Men with high-risk mHNPC treated with abiraterone had significantly longer TTCR than those treated with CAB therapy and ADT alone and were more likely to achieve a PSA level of ≤0.2. Our results provide important evidence regarding the initial treatment in men with newly diagnosed mHNPC.

## Supporting information

**S1 Data.**
(XLSX)

## Author Contributions

**Conceptualization:** Kent Kanao.

**Data curation:** Kent Kanao, Takayuki Takahashi, Yuta Umezawa, Takashi Okabe, Go Kaneko, Suguru Shirotake, Koshiro Nishimoto, Masafumi Oyama.

**Formal analysis:** Kent Kanao.

**Funding acquisition:** Kent Kanao.

**Writing – original draft:** Kent Kanao.

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
