## [Decision Letter · Decision Letter 0]

13 Jul 2022

PONE-D-22-07437

Efficacy of abiraterone acetate for high-risk hormone-naïve metastatic prostate cancer: a comparison with combined androgen blockade therapy with bicalutamide and androgen deprivation therapy alone.

PLOS ONE

Dear Dr. Kanao,

Thank you for submitting your manuscript to PLOS ONE. After careful consideration, we feel that it has merit but does not fully meet PLOS ONE’s publication criteria as it currently stands. Therefore, we invite you to submit a revised version of the manuscript that addresses the points raised during the review process.

Please note that we have only been able to secure a single reviewer to assess your manuscript. We are issuing a decision on your manuscript at this point to prevent further delays in the evaluation of your manuscript. Please be aware that the editor who handles your revised manuscript might find it necessary to invite additional reviewers to assess this work once the revised manuscript is submitted. However, we will aim to proceed on the basis of this single review if possible.

The reviewer has raised a number of concerns which requires additional attention. In particular they have requested clarification to the study design (such as the duration of patient follow up), and baseline clinicopathological data.

Could you please revise the manuscript to carefully address the concerns raised?

We look forward to receiving your revised manuscript.

Kind regards,

Lucinda Shen, MSc

Staff Editor

PLOS ONE

2. Please note that in order to use the direct billing option the corresponding author must be affiliated with the chosen institute. Please either amend your manuscript to change the affiliation or corresponding author, or email us at plosone@plos.org with a request to remove this option.

Reviewers' comments:

Reviewer's Responses to Questions

**Comments to the Author**

1. Is the manuscript technically sound, and do the data support the conclusions?

Reviewer #1: Partly

2. Has the statistical analysis been performed appropriately and rigorously? 

Reviewer #1: Yes

3. Have the authors made all data underlying the findings in their manuscript fully available?

Reviewer #1: No

4. Is the manuscript presented in an intelligible fashion and written in standard English?

Reviewer #1: Yes

5. Review Comments to the Author

Reviewer #1: The authors evaluated the treatment efficacy of abiraterone for the management of hormone naïve metastatic prostate cancer. The main outcomes were time to castration resistance (TTCR) and PSA response. Slight typographical errors in Lines 171-173 and 226.

By showing a prolonged TTCR in the ABI group, the authors have concluded that abiraterone is superior to other treatments. Without discussing the tolerability of abiraterone, the data only suggests a superiority in efficacy. For clarity, the authors should specify "superiority in treatment efficacy".

The duration of follow -up is unclear. Were any patients lost to follow-up? Were all patients followed-up uniformly for the same duration?

The clinicopathologic data shows that the groups were fairly comparable. Given the limited sample size, was the baseline testosterone level comparable between groups? If the pre-treatment testosterone data is unavailable, the authors may provide reasons why.

6. PLOS authors have the option to publish the peer review history of their article (what does this mean?). If published, this will include your full peer review and any attached files.

Reviewer #1: **Yes: **Akintunde T Orunmuyi

---

## [Author Response · Author response to Decision Letter 0]

23 Aug 2022

We would like to thank the editors and reviewer for helpful and insightful comments. We agreed with the points raised by them and revised the manuscript. 

Reviewer #1 comment 1: 

The authors evaluated the treatment efficacy of abiraterone for the management of hormone naïve metastatic prostate cancer. The main outcomes were time to castration resistance (TTCR) and PSA response. Slight typographical errors in Lines 171-173 and 226.

Response:

We thank the reviewer for noting these typographical errors. We have corrected these typographical errors.

Reviewer #1 comment 2: 

By showing a prolonged TTCR in the ABI group, the authors have concluded that abiraterone is superior to other treatments. Without discussing the tolerability of abiraterone, the data only suggests a superiority in efficacy. For clarity, the authors should specify "superiority in treatment efficacy".

Response:

We thank the reviewer for raising an important point. We reviewed the medical records of all patients and investigated adverse events. Adverse events with dose reduction or interruption occurred in 7 patients, all in the abiraterone group. Most of them were ALT/AST increase. All patients who interrupted dose resumed abiraterone and none discontinued.

We add the Table 3 which shows details of the patients in the results section and add “Adverse events with dose reduction or interruption occurred in 7 patients, all in the abiraterone group. Most of them were ALT/AST increase. All patients who interrupted dose resumed abiraterone and none discontinued. Details of adverse events of abiraterone group were shown in Table 3.” and add sentences in the discussion section as follows: “In this study, adverse events with dose reduction or interruption occurred in 7 patients, all in the abiraterone group. However, all patients who interrupted dose resumed abiraterone and none discontinued. Therefore, men with high-risk mHNPC treated with abiraterone had a manageable safety profile.”

Reviewer #1 comment 3: 

The duration of follow-up is unclear. Were any patients lost to follow-up? Were all patients followed-up uniformly for the same duration?

Response:

As described in the results section, the median (range) follow-up period in the ABI, CAB, and ADT groups was 12.6 (3–90), 9.6 (1–115), and 9.6 (3–52) months, respectively. In total, 9, 69, and 27 men in the ABI, CAB, and ADT groups, respectively, progressed to castration-resistant prostate cancer (CRPC) during the follow-up period.

We did not mention, but 13 patients were referred to another hospital during the treatment, 6 in the ABI and 7 in the CAB groups, and 3 patients died of other causes. The median follow-up period of ABI group was a little longer than CAB and ADT groups, but we followed these patients uniformly though time of treatment initiation varied.

We add a sentence in the results section as follows: “The median follow-up period of ABI group was a little longer than CAB and ADT groups, but we followed these patients uniformly though time of treatment initiation varied. Thirteen patients were referred to another hospital during the treatment, 6 in the ABI and 7 in the CAB groups, and 3 patients died of other causes.”

Reviewer #1 comment 4: 

The clinicopathologic data shows that the groups were fairly comparable. Given the limited sample size, was the baseline testosterone level comparable between groups? If the pre-treatment testosterone data is unavailable, the authors may provide reasons why.

Response:

In this study, we could not compare the pre-treatment testosterone because most of CAB and ADT patients did not evaluate pre-treatment testosterone. Recent study demonstrated that pre-treatment androgen levels are associated with OS in mCRPC patients treated with androgen synthesis inhibitors. However, to our knowledge, there is no study which demonstrated that pre-treatment androgen levels are associated with OS in mHNPC patients. In addition, the pre-treatment testosterone data is unavailable in the LATITUDE study, either. Therefore, we did not show the baseline testosterone level in this study.

---

## [Decision Letter · Decision Letter 1]

28 Sep 2022

Efficacy of abiraterone acetate for high-risk hormone-naïve metastatic prostate cancer: a comparison with combined androgen blockade therapy with bicalutamide and androgen deprivation therapy alone.

PONE-D-22-07437R1

Dear Dr. Kanao,

We’re pleased to inform you that your manuscript has been judged scientifically suitable for publication and will be formally accepted for publication once it meets all outstanding technical requirements.

Kind regards,

Donovan Anthony McGrowder, PhD., MA., MSc

Academic Editor

PLOS ONE

Additional Editor Comments (optional):

Dear Dr. Kanao,

The manuscript entitled “Efficacy of abiraterone acetate for high-risk hormone-naïve metastatic prostate cancer: a comparison with combined androgen blockade therapy with bicalutamide and androgen deprivation therapy alone” was revised in accordance with the reviewers’ comments and is provisionally accepted pending final checks for formatting and technical requirements.

Regards,

Dr. Donovan McGrowder (Academic Editor)

---

## [Editor Report · Acceptance letter]

10 Oct 2022

PONE-D-22-07437R1 

Efficacy of abiraterone acetate for high-risk hormone-naïve metastatic prostate cancer: a comparison with combined androgen blockade therapy with bicalutamide and androgen deprivation therapy alone 

Dear Dr. Kanao:

I'm pleased to inform you that your manuscript has been deemed suitable for publication in PLOS ONE. Congratulations! Your manuscript is now with our production department. 

Kind regards, 

on behalf of

Dr. Donovan Anthony McGrowder 

Academic Editor

PLOS ONE